# Comparative proteomics reveals that fatty acid metabolism is involved in myocardial adaptation to chronic hypoxic injury

Hu Chen[1]☉*, Shiran Yu[1,2]☉, Xiaoyun Zhang[3], Yujie Gao[4], Hongqi Wang[1], Yuankun Li[1], Dongsheng He[1], Weikun Jia[1]*

1 Department of Cardiothoracic Surgery, School of Clinical Medicine and The First Affiliated Hospital of Chengdu Medical College, Chengdu, China, 2 Department of Thoracic Surgery, The Third Affiliated Hospital of Chengdu Medical College, Pidu District People's Hospital, Chengdu, China, 3 Department of Cardiology, Pengzhou People's Hospital, Pengzhou, China, 4 Department of Stomatology, The First Affiliated Hospital of Chengdu Medical College, Chengdu, China

☉ These authors contributed equally to this work.
* chenhu126@126.com (HC); wk_jia315@163.com (WJ)

**Data Availability Statement:** All relevant data are within the paper and its Supporting Information files. The raw data of the proteomics are available

## Abstract

Congenital heart disease (CHD) is the most serious form of heart disease, and chronic hypoxia is the basic physiological process underlying CHD. Some patients with CHD do not undergo surgery, and thus, they remain susceptible to chronic hypoxia, suggesting that some protective mechanism might exist in CHD patients. However, the mechanism underlying myocardial adaptation to chronic hypoxia remains unclear. Proteomics was used to identify the differentially expressed proteins in cardiomyocytes cultured under hypoxia for different durations. Western blotting assays were used to verify protein expression. A Real-Time Cell Analyzer (RTCA) was used to analyze cell growth. In this study, 3881 proteins were identified by proteomics. Subsequent bioinformatics analysis revealed that proteins were enriched in regulating oxidoreductase activity. Functional similarity cluster analyses showed that chronic hypoxia resulted in proteins enrichment in the mitochondrial metabolic pathway. Further KEGG analyses found that the proteins involved in fatty acid metabolism, the TCA cycle and oxidative phosphorylation were markedly upregulated. Moreover, knockdown of CPT1A or ECI1, which is critical for fatty acid degradation, suppressed the growth of cardiomyocytes under chronic hypoxia. The results of our study revealed that chronic hypoxia activates fatty acid metabolism to maintain the growth of cardiomyocytes.

## Introduction

Congenital heart disease (CHD) is a congenital defect that affects approximately 1% of infants, and it is associated with high morbidity and mortality rates. A few decades ago, few patients with moderate or severe CHD survived to adulthood. CHD can be caused by environmental exposure to teratogens. CHD encompasses several cardiac defects that are grouped based on the character of the structural heart defect, resulting in abnormal blood flow patterns [1]. Only

at ProteomeXchange database (https://www.iprox.cn/page/project.html?id=IPX0008468000).

**Funding:** In addition, this work was supported partly by the High-level Talents Research Foundation of the First Affiliated Hospital of Chengdu Medical College (CYFY-GQ09 and CYFY-GQ37), Research Foundation of the First Affiliated Hospital of Chengdu Medical College (CYFY2019ZD04), Natural Science Foundation of Chengdu Medical College (CYZZD21-01), Sichuan Medical Research Youth Innovation Fund (Q21037) and Key Clinical Specialty Construction Project of Sichuan Province (2024GXWKP002). The funders had no role in study design, data collection and analysis, decision to publish, or preparation of the manuscript.

**Competing interests:** The authors have declared that no competing interests exist.

a few cases (approximately 15%) of CHD can be traced to a known cause. Due to ongoing innovations in diagnosis, surgery, and care, survival is now possible for most CHD patients [2].Cyanotic congenital heart disease (CCHD) is the most severe type of CHD [3]. CCHD causes special anatomical changes that inhibit the blood of the right ventricle from entering the pulmonary capillaries for full oxygenation, resulting in long-term hypoxia [2]. Children with CCHD have early onset of clinical symptoms, and some of them present with tetraplegia, convulsions, cerebrovascular accidents, and even sudden death [4]. The causes of CCHD have not been fully elucidated. An increasing amount of evidence has revealed some factors that might be linked to CCHD, including: abnormal chromosomes, genetics, and illness in the mother during pregnancy [5]. Chronic hypoxia is the basic pathophysiological process underlying CCHD [6]. Therefore, studying the mechanism of cardiomyocytes under chronic hypoxia may lead to new treatment strategies for patients with cyanotic coronary heart disease.

When oxygen levels decline, cells look for adaptive mechanisms to survive and grow. Hypoxia inducible factor 1α (HIF1α), which contains an oxygen-dependent degradation domain (ODD) [7], has been widely studied and is recognized as the key regulator of the cell defense against hypoxic stress [8]. Under normoxic conditions, HIF1α is hydroxylated on Prolines 402 and 564, two conserved residues, by prolyl hydroxylase (PHD). Hydroxylated HIF1α is bound by von Hippel-Lindau (VHL), an E3 ligase, resulting in its ubiquitination and proteasomal degradation. Under hypoxia, the enzyme activity of PHD is lost. HIF1α cannot be hydroxylated, which leads to HIF1α stabilization and dimerization with HIF1β and then subsequent transactivation of the expression of many key genes [9]. HIF-1α alters the expression of a series of proteins, for example, it upregulates vascular endothelial growth factor (VEGF) expression to induce neovascularization and improve cardiac diastolic function [10]. In addition, hypoxia or inadequate oxygen can result in protein unfolding or misfolding, thus triggering endoplasmic reticulum stress (ER stress) [11]. Proteins, as the primary components of the cell, are the executors of many important physiological processes. To adapt to ER stress, cells activate the unfolded protein response (UPR) to relieve the pressure of ER stress by reducing the synthesis of new proteins or accelerating the refolding of unfolded or misfolded proteins. The UPR is the main signaling pathway of the cellular adaptive response during endoplasmic reticulum stress. As one of the three branches of the UPR effect, activated transcription factor 6 (ATF6) plays an essential role in protein folding and maintaining cell homeostasis [12]. Notably, cardiac failure almost never occurs in infants with CCHD, who can resist the hypoxic-ischemic challenge during cardiac surgery [13]. This phenomenon suggests that chronic hypoxia could prevent cardiomyocytes from ischemic reperfusion injury. Previous studies partially elucidated the mechanism underlying chronic myocardial injury repair [14]; however, there is still a lack of overall knowledge regarding the repair mechanism of hypoxia on myocardial injury. Therefore, understanding protein alterations under hypoxia would be helpful for revealing the mechanism of chronic myocardial injury repair.

Many early examples rely only on peptide mass fingerprinting methods and lack reliable peptide sequence data; in particular, proteomic information on hypoxia is limited. Proteomics is an effective method for uncovering the mechanisms of cells during chronic hypoxia. To develop appropriate treatment strategies, further extensive protein characterization is needed. In this study, label-free quantification (LFQ) proteomics was used for relative and absolute quantification of proteins, coupled with liquid chromatography/tandem mass spectrometry (LC-MS/MS), followed by Gene Ontology (GO), and Kyoto Encyclopedia of Genes and Genomes (KEGG) pathway analyses. Proteome changes due to chronic hypoxia in mouse cardiomyocytes (HL1) in vitro were investigated in the present work. We found that chronic hypoxia leads to alterations in proteins involved in oxidoreductase activity, and mitochondria, especially fatty acid metabolism, the TCA cycle and oxidative phosphorylation. Inhibiting fatty

acid degradation-associated proteins, such as CPT1A or ECI1, significantly suppresses myocardial cell proliferation.

## Materials and methods

### Cell culture and hypoxia treatment

Mouse cardiomyocytes HL1 were a kind gift from Professor Jie Chao, and cultured in a Claycomb medium (NO. 51800C, Sigma-Aldrich, Shanghai, China) containing 10% fetal bovine serum (Hyclone) and 100 U/mL penicillin and 100 μg/mL streptomycin (GIBCO, Rockville, MD, USA), Norepinephrine 10 nM (NO. A0937, Sigma-Aldrich) and L-Glutamine 2 mM (NO. A7506, Sigma-Aldrich). HEK-293T cell line were obtained from the American Type Culture Collection (ATCC) and cultured in DMEM (Gibco, Shanghai, China) contained 10% fetal bovine serum. All cell lines were cultured at 37°C in a humidified incubator under 5% $CO_2$. To induce hypoxia, HL1 cells were delivered into an anaerobic chamber maintained in humidified atmosphere of 5% $CO_2$, 10% $O_2$, and 85% $N_2$ at 37°C [15]. In the anaerobic chamber, the medium was replaced with Claycomb medium which had been saturated with $N_2$ gas for 40 min. Cells were treated for 0h, 24h, 48h or 72h, and then subjects to follow-up analysis. For cell viability analysis, cells were digested and subjected to trypan blue exclusion assay (C0011, Beyotime, China) according to the manufacturer's instructions.

### Protein extraction and peptide enzymatic hydrolysis

HL1 cells treated with hypoxia were grinded as powder with liquid nitrogen and dissolved in lysis buffer (8M urea, 2mM EDTA, 10mM DTT and 1% PMSF), then ultrasonicated on ice for 5 min. The lysates were centrifuged at 12000g for 15 min at 4°C. The supernatants were transferred to a clean tube. Protein concentration was measured by Bradford Protein Assay Kit (P0006, Beyotime, China) [16]. Each sample were reduced with 10 mM DTT for 0.5 h at 37°C, and then alkylated with sufficient iodoacetamide for 0.5 h at room temperature in the dark. Samples were mixed with precooled acetone (4 times volume) and incubated at -20°C for 2 h. Then samples were centrifuged. And the precipitation was collected. After washing twice with cold acetone, the pellet was dissolved in lysis buffer (8 M urea, 100 mM TEAB). Protein concentration was measured again by Bradford Protein Assay Kit.

Each sample containing protein 100 μg was reduced with DTT at 37°C for 60 min firstly, then alkylated for 45 min in darkness at 25°C. 100 mM TEAB was added to reduce the concentration of urea to 2 M. Finally, pooled protein samples were digested with Trypsin Gold (protein: trypsin = 50:1, Sigma, USA) at 37°C overnight. The single-sample was all desalted with C18 cartridge to remove the high urea, and desalted samples were dried by vacuum centrifugation.

### HPLC fractionation and LC-MS/MS analysis

High performance liquid chromatography (HPLC) off-line peptide separation (PS) technique has been used to enhance the detection of proteins by liquid chromatograph-tandem mass spectrometry (LC-MS /MS) and is a more effective method to improve the coverage of MS proteome [17]. The dried peptides were reconstituted with high-pH reverse-phase HPLC solution A (2% ACN, pH 10) and fractionated by using a C18 column (Waters BEH C18 4.6 × 250 mm, 5 μm) on a Shimadzu LC 15 HPLC operating at 0.6 mL/min. Mobile phases A (2% ACN, adjusted pH to 10.0 using ammonium hydroxide) and B (98% ACN, adjusted pH to 10.0 using ammonium hydroxide) were used to develop a gradient elution. The solvent gradient was set as follows: 0–5 min, 0% B; 5–55 min, 0–30% B; 55–67 min, 30%-100% B; 67–75 min, 100% B;

75–78 min, 100%-0% B; 78–88 min, 100% A. The eluates were monitored at UV 214 nm, collected for a tube per minute and merged into 4 fractions finally. All fractions were dried under vacuum and reconstituted in 0.1% (v/v) formic acid (FA) in water.

Dissolve each component sample in 20 μL of 2% acetonitrile, inject 1 μL, and then enter Q Exactive HF combined MS for identification through gradient elution of the analytical column. The samples were analyzed using the ultimate 3000 nano upgraded HPLC system equipped with Acclaim PepMap® 100 C18, 3μm, 100Å (75μm × 2 cm) trap column and Acclaim PepMap® RSLC C18, 2μm, 100Å (50 μm × 15 cm) Analytical column. The liquid flow rate is 300 mL/min. 0–10 min, 5% (buffer B: 0% acetonitrile, 0.1% FA); 10–70 min, 50–45% B; 70–73 min, 45–90%B; 73–78 min, 90% B; 78–78.1 min, 90% - 5% B, 78.1–90 min, 5% B. The eluent was sprayed via NSI source at the 2.5 kV electrospray voltage and then analyzed by MS/MS in Q Exactive HF. The mass spectrometer was operated in a data-dependent mode. Analyzer: Orbitrap; Scan Type: Full; Resolution: 60000; Polarity: +; Scan Range: 400–2000 m/z. Analyzer: Orbitrap; Resolution: 15000; Dynamic Exclusion: 30 s; Charge state: Reject 1; Current Scan Event: Top 15 peaks; Activation Type: HCD; Normalized Collision Energy: 27.

## Parallel reaction monitoring analysis

To verify the reliability of the proteomics, a label-free targeted parallel reaction monitoring (PRM) method was used [18]. Parallel reaction monitoring (PRM) is an ion monitoring technique based on high-resolution and high-precision mass spectrometry. The principle of this technique is comparable to SRM/MRM, but it is more convenient in assay development for absolute quantification of proteins and peptides. It is most suitable for quantification of multiple proteins in complex sample with an attomole-level detection. PRM technology is a novel technology for verification of antibody-free proteins, which is used for verification of relative quantitative proteomic data such as label free, iTRAQ, TMT, SILAC, or for targeted quantitative analysis of target proteins in complex samples. PRM technology not only has the SRM/MRM target quantitative analysis capabilities, but also have the qualitative ability. (1) The mass accuracy can reach to ppm level, which can eliminate the background interference and false positive better than SRM/MRM, and improve the detection limit and sensitivity in complex background effectively; (2) Full scan of product ions, without the need to select the ion pair and optimize the fragmentation energy, easier to establish the assay; (3) a wider linear range: increased to 5–6 orders of magnitude [19]. In this study, eighteen differentially expressed proteins (DEPs), including one internal standard housekeeping protein were chosen to verify the reliability of the label-based proteomics. Peak areas were extracted from PRM mass spectrum data using Skyline software.

## Data analysis

The result of MS/MS raw data were searched against the reviewed *Mus musculus* Proteome database downloaded from Uniport database using SEQUEST software integrated in MaxQuant 1.6.6.0. Trypsin was chosen as enzyme and two missed cleavages were allowed. Carbamidomethylation was set as a fixed modification, oxidation, and acetylation in N-terminal were set as variable modification. The quantitative method was set to LFQ. The searches were performed using a peptide mass tolerance of 20 ppm/0.02 Da after recalibration for precursor masses and product ion tolerance of 20 ppm, peptide- and protein-level false discovery rate (FDA) thresholds in the "target-decoy" database resulting in 1% in each biological replicate. The principal component analysis was performed by using the scatterplot function from seaborn 0.11.2 Python library [20]

Then proteins were classified by gene ontology annotation based on three categories: biological process, cellular component, and molecular function. GO annotation proteome was derived from the UniProt-GOA database (http://www.ebi.ac.uk/GOA/) [21]. KEGG Pathway is part of Kyoto Encyclopedia of Genes and Genomes database [22–24] was used to annotate protein pathway. KEGG online service tools KAAS were used to annotate protein's KEGG database description and then map the annotation result onto the KEGG pathway database via KEGG mapper, KEGG online service tools. InterProScan [25], a sequence analysis application was also used for protein domain annotation based on protein sequence alignment algorithm and the InterPro domain database. Expression-based Clustering for different protein groups were used to explore potential relationships between different protein groups at special protein function (such as KEGG Pathway). Firstly, all the protein groups obtained after functional enrichments analysis along with their $P$ values were collated. Secondly, those categories enriched in one of the protein groups with a $P$ value $<0.05$ were sorted. This filtered $P$ value matrix was transformed by the function x = −log10 (P value). Thirdly, z-transformed applies on x values for each functional category and z scores were clustered by one-way hierarchical clustering (Euclidean distance, average linkage clustering). Finally, cluster membership was visualized by a heatmap using the "heatmap" function from the R-package.

## Plasmid construction and generation of stable cell lines

shRNA oligos targeting to CPT1A and ECI1 were cloned into pLKO.1 vector according to protocol from Addgene instruction. The shRNA-targeting sequences for mouse CPT1A and ECI1 are following: shCPT1A-#1: 5'-GCTATGGTGTTTCCTACATTA-3', shCPT1A-#2: 5'-ATGGACTCTAGTGATACAAAC-3'; shECI1-#1: 5'- GATGACCAAGTGGTTAGCTAT -3', shECI1-#2: 5'- CCAGTTTCATCTCCAAAGACT -3'; All plasmids used in this study were confirmed through DNA sequencing.

Lentiviral particles were prepared as previously study [26]. For the generation of stable cell lines, cells were infected with recombinant lentiviruses, then treated with 2 μg/mL puromycin (Sigma Inc, USA) at 48 h of post-infection for generation of stable cells expressing desired gene or shRNA.

## Western blotting

Western blot analysis was used to detect protein expression and was performed as described previously [27]. Antibody used in this study were as follows: CPT1A (1:500, ab228964, Abcam, Shanghai, China); ECI1 (1:1000, ab186752, Abcam); HIF1α (1:500, ab307829, Abcam); Actin (1:3000, cat. 250132, ZEN-BIOSCIENCE, China).

## Immunofluorescent

Immunofluorescent analysis was performed as following steps. Cells grown on coverslips were fixed with 4% polyformaldehyde in PBS, permeabilized with 0.1% Triton X-100 in PBS, blocked with 4% bovine serum albumin in PBS, hybridized with rabbit-anti-CD36 (ab252922, abcam, 1:100), and Rhodamine (TRITC)-conjugated donkey-anti-rabbit IgG (711-025-152, Jackson ImmunoRsearch, PA, USA) and counter-staining with DAPI (Beyotime) for subsequent detection. Coverslips were mounted with ProLong Gold antifade reagent (Invitrogen). Images were acquired using the Leica TCS SP5 II system. The free software Image J. Fiji coupled with MorphoLibJ plugin was used to quantify the protein level of CD36 on the cell membrane and in the total cell.

### Fatty acids assay

HL1 cells were cultured under normoxic and hypoxic condition for 72h, then lysed for determination of fatty acids by Free Fatty Acid Quantitation Kit (MAK044, Sigma-Aldrich). The amounts of free fatty acids were normalized to the content of total cellular protein which measured by BCA Protein Assay Kit (Cat. No. 23227, Pierce, Shanghai, China).

### RTCA

To test cell proliferation, HL1 cells were seeded at density of 5000 cells/well in Real-Time Cell Analyzer (RTCA, ACEA Biosciences, Inc., CA, USA), then cultured in medium supplemented with oleic acid (5μM) at 37˚C with humidified atmosphere of 5% $CO_2$, 10% $O_2$, and 85% $N_2$, the cell index signals were recorded every 1 hour for 80 hours.

### Statistical analysis

Results are presented as mean ± standard deviation (SD). Statistical analysis was calculated by using Student's t-test and P value $< 0.05$ was considered as statistical significance.

## Results

### Experimental design and quality control

Some patients with CHD do not undergo surgery and remain susceptible to chronic hypoxia, thus suggesting that some protective mechanism might exist in CHD patients. To investigate the protective mechanism of cardiomyocytes under chronic hypoxia, we cultured mouse cardiomyocytes (HL1) under 10% $O_2$, a moderate hypoxic state, for different times (Fig 1A) and analyzed the viability of the cells. As shown in Fig 1B, hypoxia (10% $O_2$) slightly reduced cell viability; however, there were no significant differences in cell viability between hypoxia and normoxia or between different hypoxia treatments. Then the cells were subjected to LC-MS/ MS analysis to detect the expression of different proteins. To determine the heterogeneity among the cells suffering from hypoxia treated for different times, we performed principal component analysis (PCA). The PCA indicated that the proteins derived from cells treated by hypoxia for 0h and 72h, 24h or 48h and 72h are heterogeneous and show significant variation in protein levels across cells (Fig 1C). Peptides were identified according to both their molecular weight (MW) and ion fragmentations (Fig 1D). Moreover, the distribution of mass error was close to zero and most were less than 4 peptide mass errors (PPMs), indicating the high accuracy of modified peptide data obtained by MS (Fig 1E). The repeatability analysis showed a considerably high correlation between biological replicates, demonstrating the reliability of our experimental protocol (Fig 1F). Parallel reaction monitoring (PRM) quantitative strategy analysis was carried out to verify the accuracy and repeatability of the proteomic analysis results, and eleven proteins that may play an important role in chronic hypoxia were selected. The verification results of all proteins, peptides, and measured peptide peak areas are shown (S1 Table and S1 Fig). The quantitative results of PRM of eleven proteins distributed in each time period are consistent with the proteomics results, indicating that the label-free proteomics results are reliable.

### Protein identification and differentially expressed protein screening

In this study, 3881 proteins were identified in three replicates of the LC-MS/MS experiments with a high confidence of peptide selection (FDR = 0.01). Among them, 2607 proteins achieved quantitative significance, and all the proteins were completely annotated with the accession number for each identified protein arranged by GO terms and KEGG pathway

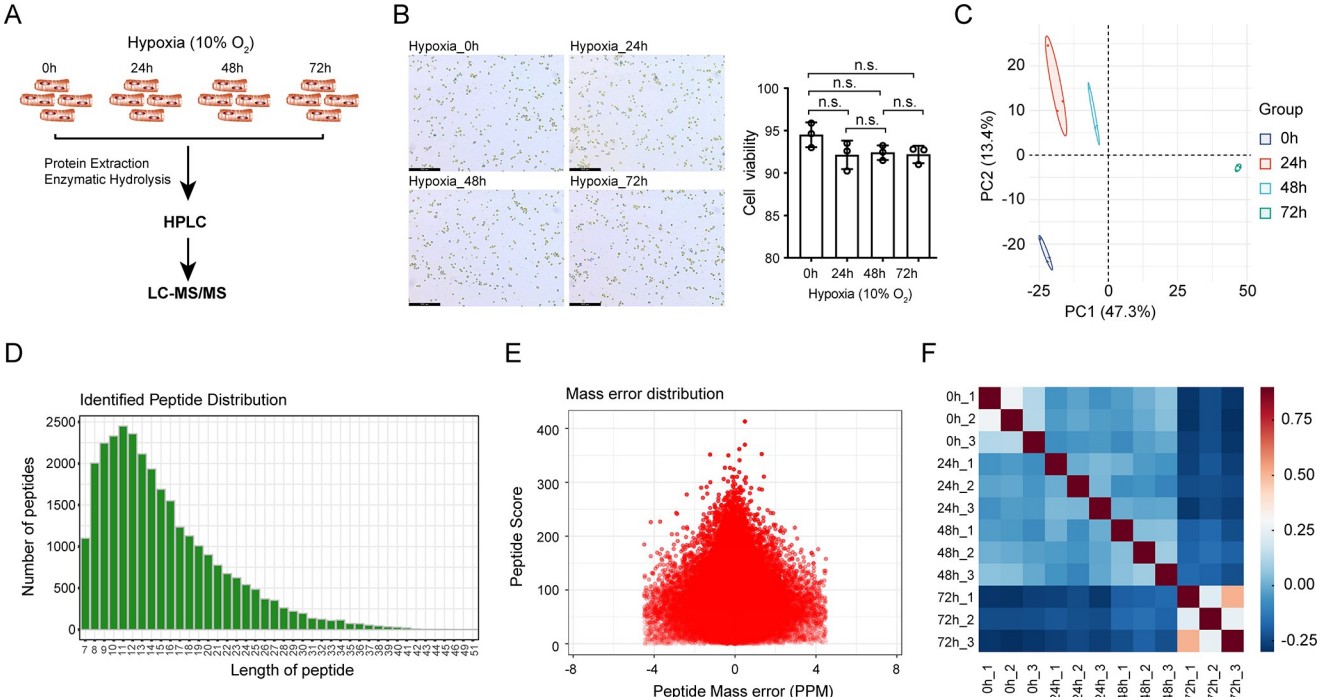

**Fig 1. Study design and repeatability analyses.** (A) Workflow plot of the study design. (B) Cell viability was measured by trypan blue exclusion assay at different time points after hypoxia treatment. Images (left) and quantitative data (right) are presented, n.s. stands for not significant. (C) PCA plot of HL1 cells in response to hypoxia treatment for different time. (D) Distribution of identified peptide length. (E) The distribution of mass error indicates a high accuracy of modified peptide data obtained from MS. (F) Spearman's correlation analysis showed the correlation of differentially expressed proteins (log2 ratio) between biological replicates.

(S1 Table). Our results revealed that differentially expressed proteins were significantly expressed among the four treatments. Compared with the cells treated with hypoxia for 0 h, there were 76 proteins with significant differential expression (fold change, FC > 1.5, $p$ value < 0.01) after hypoxia treatment for 24 h; 68 proteins were upregulated and 49 proteins were downregulated. When comparing 48 h vs. 24 h, there were 34 upregulated proteins and 42 downregulated proteins. A total of 631 proteins changed dynamically between 72 h and 48 h, including 315 upregulated and 316 downregulated proteins (Fig 2A).

Based on the results of the second level (Level 2), we found that differentially expressed proteins (DEPs) at different times are involved in many biological processes, especially metabolic processes. Furthermore, molecular function enrichment showed that many DEPs were enriched in catalytic activity (Fig 2B), indicating that hypoxia might regulate the metabolic process of HL1 cell by affecting the catalytic activity of enzymes.

## Hypoxia leads to oxidoreduction alterations in HL1 cells

To better understand the functions of the DEPs in regulating HL1 cells after hypoxia treatment, GO ontology was performed. As shown in Fig 3A–3C, GO analysis of molecular function showed that proteins involved in the regulation of oxidoreductase activity were significantly changed after hypoxia treatment. Furthermore, we analyzed the expression of proteins involved in oxidoreductase activity, and found that AL1L2 (mitochondrial 10-formyl-tetrahydrofolate dehydrogenase), AOFA (amine oxidase [flavin-containing] A), IDH3B (isocitrate dehydrogenase [NAD] subunit, mitochondrial), PGH2 (prostaglandin G/H synthase 2), and NCP (NADPH-cytochrome P450 reductase) gradually increased with the time course of

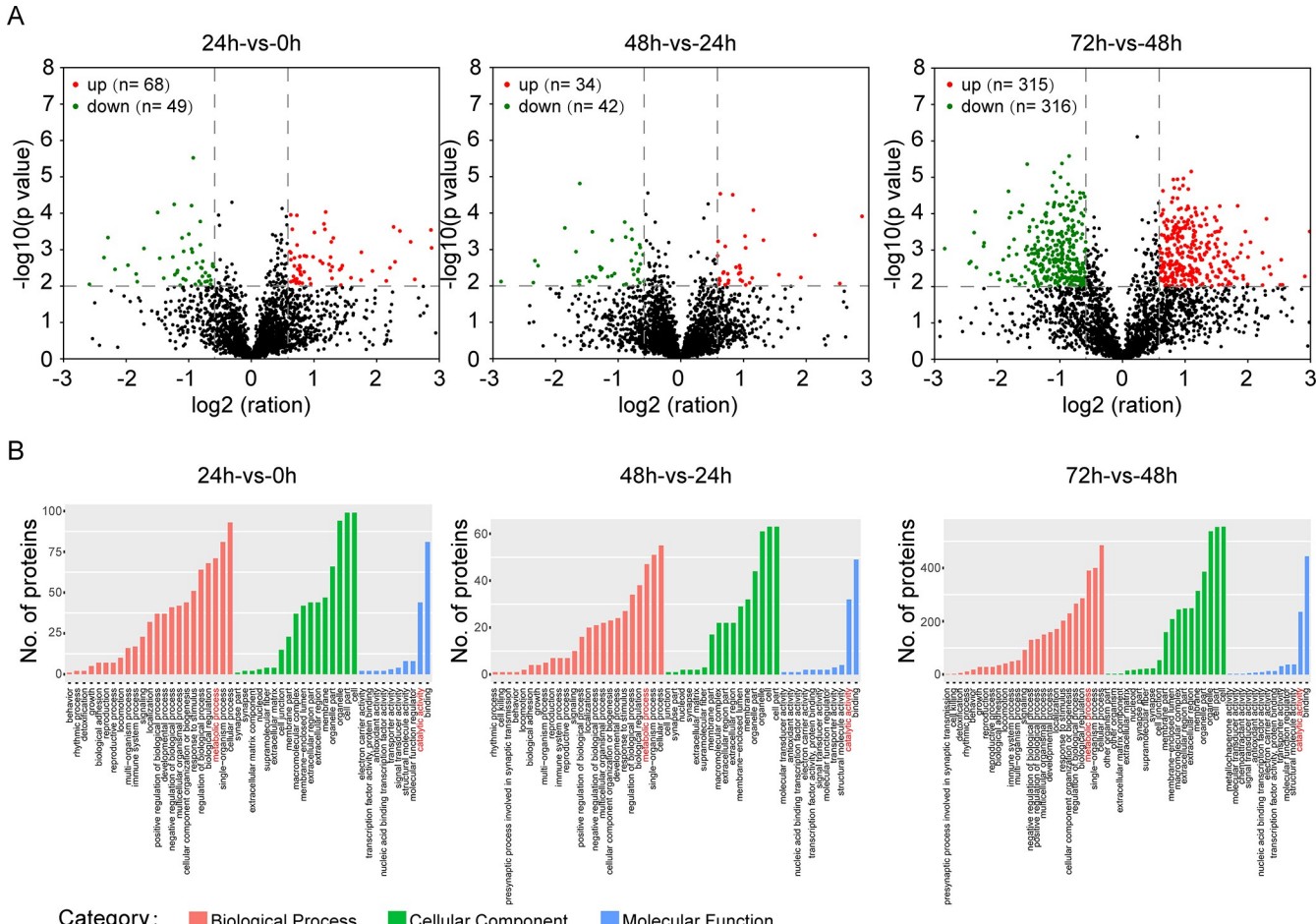

**Fig 2. DEP analyses and gene ontology classification.** (A) Differentially expressed protein (DEP) annotation. (B) Histogram of gene ontology classification. The results are summarized in three main categories: biological process, cellular component and molecular function. The right y-axis indicates the number of genes in a category. The left y-axis indicates the percentage of a specific category of genes in that main category.

hypoxia treatment (Fig 3D). Conversely, NDUA7 (NADH dehydrogenase [ubiquinone] 1 alpha subcomplex subunit 7) gradually decreased with the time of hypoxia treatment. ALD (aldo-keto reductase 1 member B1), CB3 (carbonyl reductase 3), and PDX6 (peoxiedoxin-6) also increased continuously. DOHH (deoxyhypusine hydroxylase), DY (dihydrofolate reductase), ferritin and HMOX1 (heme oxygenase 1) only increased after 24 h of hypoxia treatment. DHS1 (dehydrogenase/reductase SD family member 1) and SH3L3 (SH3 domain-binding glutamic acid-ich-like protein 3) significantly decreased after 24 h of hypoxia treatment. These data indicate that chronic hypoxia might regulate oxidoreductase activity by changing the protein expression of oxidoreductase.

## Chronic hypoxia alters the mitochondrial metabolic pathway

Although hypoxia affects the protein expression of oxidoreductase, chronic hypoxia is a state of long-term oxygen insufficiency and might alter the function of metabolism. To investigate which function was affected by chronic hypoxia, we analyzed the DEPs between 48 h and 72 h. During this time period, the cells experienced long-term hypoxia treatment, and the highest number of DEPs were also found. As shown in Fig 4, mitochondrion-associated proteins,

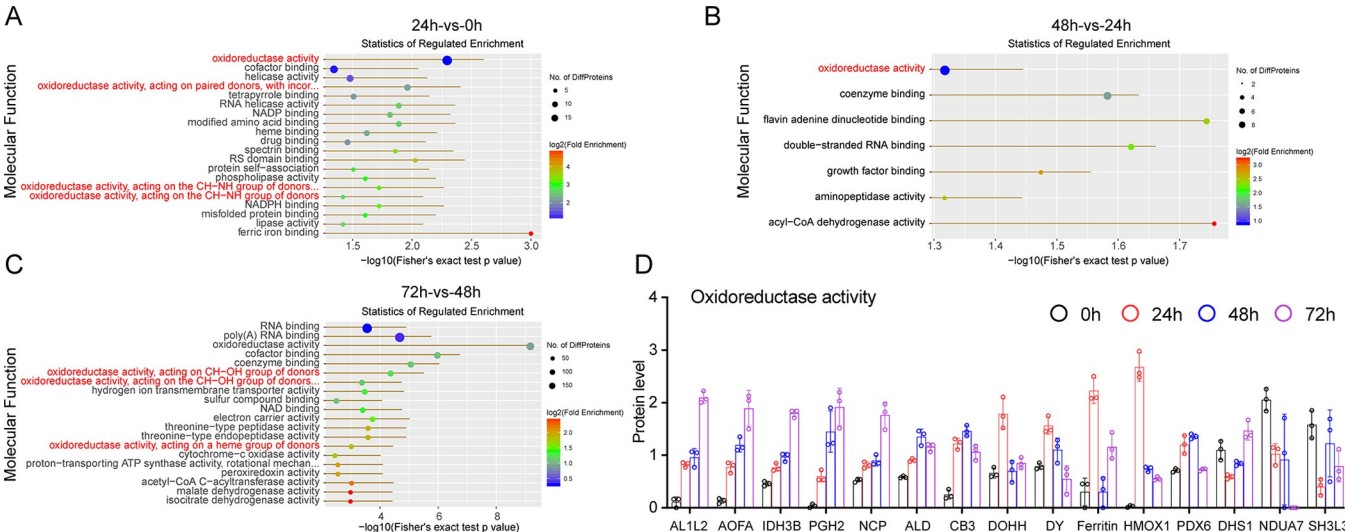

**Fig 3. Visualization of significantly enriched GO terms in HL1 in response to hypoxia.** A) DEPs were enriched according to biological processes (A), cellular components (B), and molecular functions (C). DEPs involved in oxidoreductase activity are presented (D).

mitochondrial nucleoid-associated proteins, mitochondrial inner membrane-associated proteins and oxidoreductase complex-associated proteins were enriched in the functional similarity cluster Gene Ontology analysis. As expected, we found that proteins associated with the mitochondrial inner membrane were significantly enriched after 72 h of hypoxia treatment compared with 48 h of hypoxia treatment. Thus, cardiomyocytes might change the metabolic pathways of mitochondria to adapt to chronic hypoxia.

To further investigate which pathway was altered in cardiomyocytes after long-term hypoxia treatment, we performed KEGG pathway analysis. As shown in Fig 5A, the proteins involved in fatty acid degradation, fatty acid metabolism, carbon metabolism, metabolic pathways, citrate cycle (TCA cycle) and oxidative phosphorylation were significantly enriched after hypoxia treatment for 72 h relative to 48 h or 0 h. Under normoxic conditions, the heart depends predominantly (~60–90%) on fatty acid (FA) oxidation to fuel ATP generation [28]. As shown in Fig 5B, we found that the following proteins were enriched: proteins involved in fatty acid metabolism, such as ACSL4 (acyl-CoA synthetase long-chain family member 4), which catalyze the conversion of long-chain fatty acids to their active form acyl-CoA for both cellular lipid synthesis and β-oxidation; proteins involved in shuttling long-chain fatty acids into mitochondria, such as CPT1A (carnitine O-palmitoyltransferase 1 A) and CPT2 (mitochondrial carnitine O-palmitoyltransferase 2); proteins involved in catalyzing the initial step of mitochondrial β-oxidation, such as ACADL (mitochondrial long-chain specific acyl-CoA dehydrogenase), ACADM (medium-chain specific acyl-CoA dehydrogenase) and ACADV (mitochondrial very long-chain specific acyl-CoA dehydrogenase); proteins involved in the trifunctional enzyme, such as ECHA (mitochondrial trifunctional enzyme subunit-alpha) or ECHB (mitochondrial trifunctional enzyme subunit beta), which catalyzes the last three steps of mitochondrial beta-oxidation of long-chain fatty acids; and proteins involved in catalyzing the last step of mitochondrial β-oxidation, such as THIKA (peroxisomal 3-ketoacyl-CoA thiolase A), THIL (mitochondrial acetyl-CoA acetyltransferase 1) and THIM (mitochondrial 3-ketoacyl-CoA thiolase). Interestingly, DHB12 (very-long-chain 3-oxoacyl-CoA reductase), which is involved in fatty acid elongation, was also upregulated after hypoxia treatment, suggesting that chronic hypoxia might trigger the synthesis of cellular lipids in addition to

## Gene Ontology functional similarity cluster

**Fig 4. A treemap overview of significant GO biological processes in hypoxia treated HL1 cells.** The diagram shows that HL1 cells exposed to prolonged hypoxic conditions (72 h vs. 48 h) had a significant effect on proteins associated with mitochondria, the mitochondrial inner membrane, and the oxidoreductase complex. The relative sizes of the treemap boxes are based on the |log10(p value)| of the respective GO term, related terms are visualized with the same color, the color represents the significant p value of this kind of term after -log10.

oxidation. β-Oxidation cleaves fatty acids into acetyl-CoA, which eventually enters the TCA cycle and oxidative phosphorylation to synthesize ATP. As expected, chronic hypoxia also upregulated proteins associated with the TCA cycle and oxidative phosphorylation (Fig 5C and 5D). These data suggest that chronic hypoxia maintains the function of cardiomyocytes by altering the metabolic pathway.

## Hypoxia maintains HL1 cell viability by accelerating fatty acid degradation

To verify the accuracy of protein quantitation, we tested protein expression by western blotting. As expected, hypoxia significantly upregulated the protein level of HIF1α (Fig 6A). Additionally, we found that CPT1A, the rate-limiting enzyme of fatty acid β-oxidation, and ECI1, an auxiliary enzyme involved in unsaturated fatty acid oxidation, also increased markedly. To

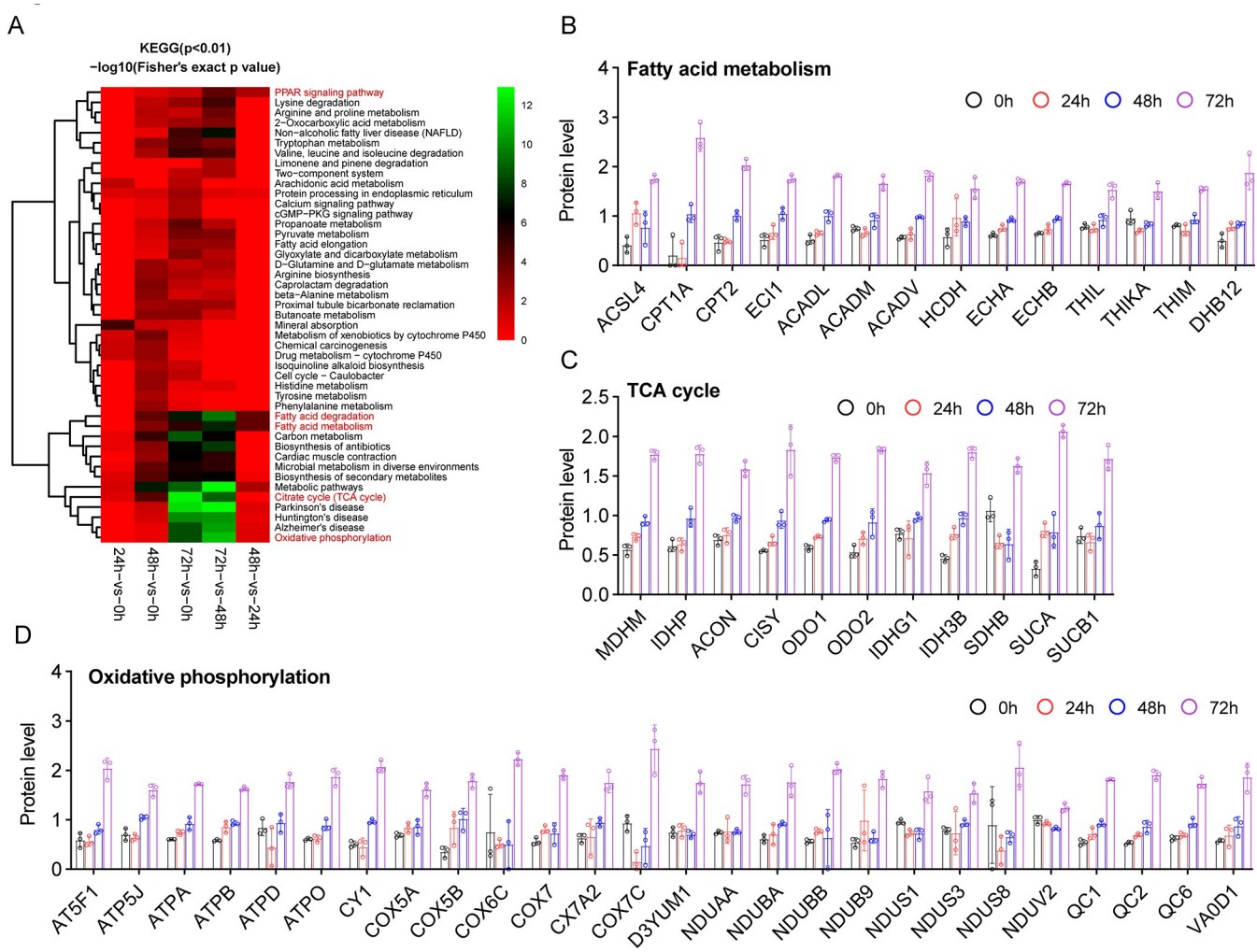

**Fig 5. KEGG enrichment and DEP representation for significant pathways.** A) KEGG enrichment-based clustering for DEPs between hypoxia treatment for 48 h and 72 h. DEPs involved in fatty acid metabolism (B), the TCA cycle (C), and oxidative phosphorylation (D) are presented.

investigate whether fatty acid β-oxidation is critical for maintaining the viability of myocardial cells under hypoxia, we first detected the content of free fatty acids under normoxic or hypoxic conditions. As shown in Fig 6B, the content of fatty acids decreased only slightly after hypoxia treatment compared with the content of fatty acids under normoxia. It is well-known that the uptake of fatty acids is also important for fatty acid degradation. However, the protein level of CD36, which is critical for fatty acids uptake [29], did not change significantly in response to hypoxia (Fig 6C–6E), indicating that chronic hypoxia might not affect fatty acid uptake. Given culture medium contained a little of fatty acids, we cultured HL1 cells under hypoxia by using medium supplemented with oleic acid. As shown in Fig 6F, the fatty acids concentration in cellular was reduced significantly after hypoxia treatment, suggesting that hypoxia might trigger fatty acids β-oxidation. Furthermore, we knocked down the expression of CPT1A or ECI1 and tested the cell growth under hypoxia. As shown in Fig 6G, two different shRNAs efficiently knocked down the protein levels of CPT1A and ECI1. RTCA experiments demonstrated that knockdown of CPT1A or ECI1 markedly inhibited cell growth (Fig 6H). These data indicate that chronic hypoxia might activate the degradation of fatty acids to maintain cardiomyocyte growth.

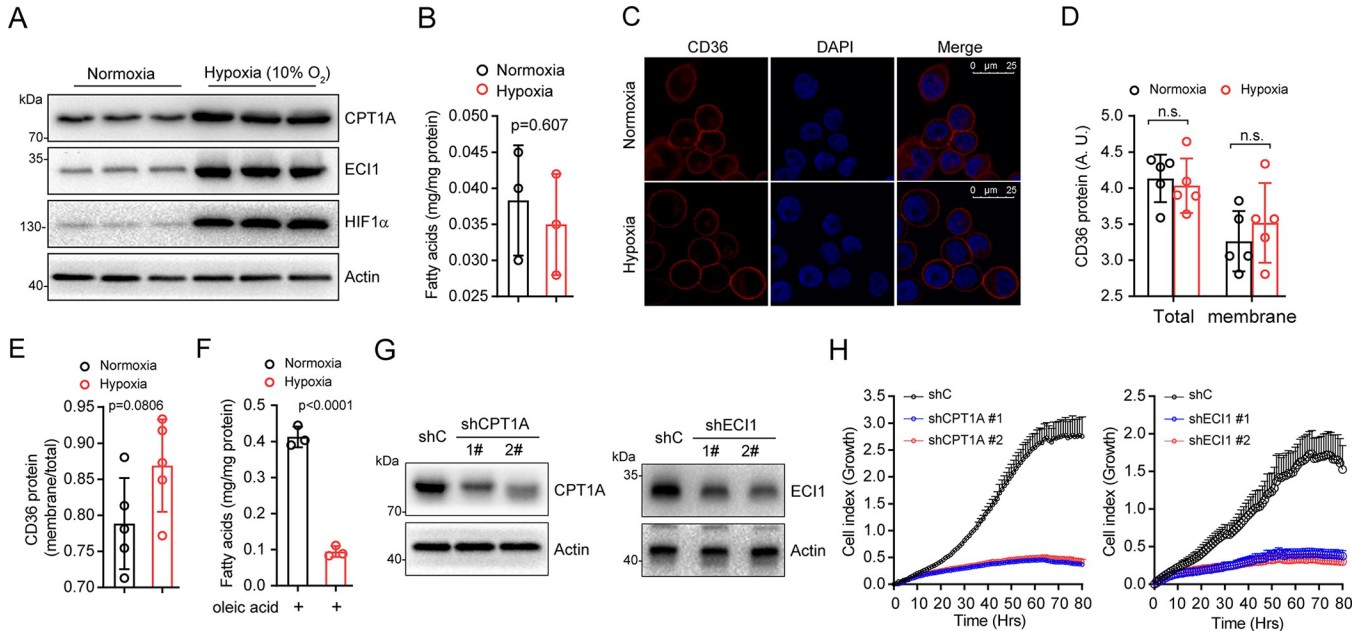

**Fig 6. Hypoxia maintains cell growth by activating the degradation of fatty acids.** HL1 cells were treated under hypoxia for 72 h. The whole cell lysates were subjected to WB analysis (A), fatty acids measurement (B) and immunofluorescent analysis (C). The protein levels of CD36 on the cell membrane or in the total cell were quantified on the base of immunofluorescent analysis (D), A.U. stands for arbitrary unit. The membrane/total ratio of CD36 were calculated (E). HL1 cells were culture in medium supplemented with oleic acid treated with hypoxia for 72 h, and followed by measuring the concentration of fatty acids (F). HL1 cells, which were infected with lentivirus encoding shRNA targeting CPT1A or ECI 1. Cells were cultured in medium supplemented with oleic acid treated with hypoxia for 72 h. The whole cell lysates were subjected to WB analysis (G) or RTCA analysis (H).

## Discussion

Cyanotic congenital heart disease (CCHD) is the most serious congenital heart disease and a complex pathophysiological condition with chronic hypoxia. For multiple potential reasons, some patients with CCHD do not undergo surgery, and they remain chronically hypoxic during their lives [14]. This phenomenon indicates that there is a protective mechanism for CCHD patients with inadequate oxygenation. Under chronic hypoxia, cardiomyocytes generate a series of compensatory changes in biochemical and metabolic processes, alter the consumption of oxygen and energy, and rebuild the homeostasis system of the cells [30, 31]. Recently, an integrated multi-omics study was performed to characterize the congenital heart and found congenital heart disease (CHD)-specific cell states in cardiomyocytes, which provided evidence of insulin resistance and induced gene expression by FOXO and CRIM1 [32]. It is well documented that HIF1α play an important role for cell in response to hypoxia. Under normoxia, HIF1α is hydroxylated by prolyl hydroxylases on prolines 402 and 564, which promotes HIF1α binding to the von Hippel–Lindau (VHL) E3 ligase, resulting in its subsequent ubiquitylation and proteasomal degradation [33]. Using proteomics to identify the key proteins in the process of chronic hypoxia can provide a deep understanding of the damage or protective mechanism of myocardial cells, which can provide new treatment strategies for patients with CCHD. To understand the mechanism, we used mild and long-term hypoxic conditions to mimic chronic hypoxia and found differentially expressed proteins enriched in the metabolic pathway of mitochondria.

The mitochondrion, an important cellular organ in eukaryotic cells, is the energy factory for cells. In addition, mitochondria also generate intermediate metabolites that are critical for anabolic pathways, such as lipid synthesis, amino acid synthesis, and nucleic acid synthesis. As

an energy reservoir, mitochondria are sensitive to hypoxia. Studies have shown that chronic hypoxia inhibits mitochondrial DNA replication and the transcription and translation of metabolism-related enzymes [34]. Mitochondrial morphology, number, respiration and complex IV enzyme activity were altered when exposed to chronic hypoxia, suggesting that hypoxia facilitates tissue diffusion of intracellular oxygen by increasing the mass and number of mitochondria in the liver and heart, thereby improving intracellular oxygen diffusion through intrahepatic mitochondrial distribution. Heart failure was related to the downregulation of mitochondrial proteins by two-dimensional electrophoresis [35]. However, we found that mitochondrion associated proteins were upregulated in this study, which was the reason that we used mild long-term hypoxic (10% $O_2$, 72 h) culture conditions. Furthermore, we found that inner membrane-associated proteins were enriched after hypoxia treatment. It is well known that many enzymes, involved in catalyzing energy production, are located in the inner membrane of mitochondria. These data indicate that myocardial cells might increase energy production to adapt to the hypoxia we used in this study.

The common pathway that mitochondria are responsible for is the TCA cycle with oxidative phosphorylation. It has been reported that cardiomyocytes are widely known to preferentially use glucose over fatty acids as energy resources in response to chronic hypoxia [14, 36]. In this study, we found 37 proteins participated into glycolysis/gluconeogenesis present varying degree increase or decline (S2 Table). And the tricarboxylic acid cycle and oxidative phosphorylation is highly enriched among all quantified proteins annotating pathways (Fig 5C and 5D). These results indicate that glucose is an important energy resource for cell survival in response to chronic hypoxia. Besides, the heart needs fatty acids to maintain its activity. We also found that mild long-term hypoxia led to significant upregulation of proteins involved in fatty acid metabolism, oxidoreductase activity, the TCA cycle and oxidative phosphorylation. Of note, the proteins were involved in fatty acid mobilization (ACSL4), transportation (CPT1A and CPT2), and mitochondrial β-oxidation (ACADL, ACADM, ACADV, ECHA, ECHB, THIKA, THIL and THIM). CPT1A, the rate-limiting enzyme in the fatty acid β-oxidation, was regulated in multiple levels. PPARα and PGC1α could enhance CPT1A expression at transcriptional level. miR-33a/b, miR-124, miR-328-3p, or miR-370 inhibits the translation of CPT1A to suppress fatty acid β-oxidation [37]. In addition, the uptake of fatty acids is also important for fatty acid degradation. CD36 is critical for fatty acid uptake [29]. However, we found that the protein level of CD36 was not changed in this study, indicating that chronic hypoxia might not affect fatty acid uptake.

Fatty acids are the primary energy resource under normoxia. The adaptation mechanism to chronic hypoxia is critical for the survival and growth of cardiomyocytes. In this study, we found that proteins involved in mitochondria, particularly in fatty acid degradation, were significantly upregulated after chronic hypoxia treatment. Furthermore, we demonstrated that chronic hypoxia could effectively upregulate the protein level of HIF-1α in line with the increase in the protein levels of CPT1A and ECI1. Knockdown of CPT1A or ECI1 dramatically inhibited the growth of myocardial cells, indicating that fatty acid oxidation is also important for cell growth.

This study had several limitations. First, we did not verify our conclusions in vivo by using an animal model because animal models of congenital heart disease are limited and have high hardware conditions. Second, the study was based on proteomics using mouse cardiomyocytes, which could not faithfully recapitulate the in vivo situation. Therefore, a future study is necessary to examine the protective role of fatty acids in vivo.

## Supporting information

**S1 File. Cell line authentication report.**
(PDF)

**S1 Fig. PRM for the abundance of the brain-specific angiogenesis inhibitor 1-associated protein 2 (B1AZ46) was determined.**
(DOCX)

**S1 Table. Putative annotations of differentially expressed proteins (DEPs) of mouse cardiac myocytes after treated with hypoxia.**
(XLSX)

**S2 Table. DEPs of glycolysis or gluconeogenesis.**
(XLSX)

**S1 Raw images. Original western blots.**
(PDF)

## Acknowledgments

The authors would like to thank Ms. Tao Li for his excellent technical assistance.

## Author Contributions

**Conceptualization:** Hu Chen.

**Data curation:** Shiran Yu, Xiaoyun Zhang.

**Formal analysis:** Xiaoyun Zhang, Yujie Gao, Hongqi Wang, Yuankun Li, Dongsheng He.

**Funding acquisition:** Yujie Gao, Weikun Jia.

**Methodology:** Hongqi Wang, Yuankun Li, Dongsheng He.

**Supervision:** Hu Chen.

**Writing – review & editing:** Hu Chen, Weikun Jia.

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
