## [Decision Letter · Decision Letter 0]

29 Jan 2024

PONE-D-23-43179Comparative proteomics reveals that fatty acid metabolism is involved in myocardial adaptation to chronic hypoxic injuryPLOS ONE

Dear Dr. Chen,

Thank you for submitting your manuscript to PLOS ONE. After careful consideration, we feel that it has merit but does not fully meet PLOS ONE’s publication criteria as it currently stands. Therefore, we invite you to submit a revised version of the manuscript that addresses the points raised during the review process.

Your manuscript has been reviewed by two experts and we received mixed recommendation. Pleasea ddress all comments as appropriate.

We look forward to receiving your revised manuscript.

Kind regards,

Partha Mukhopadhyay, Ph.D.

Section Editor

PLOS ONE

Journal Requirements:

"the High-level Talents Research Foundation of the First Affiliated Hospital of Chengdu Medical College (CYFY-GQ09 and CYFY-GQ37), Research Foundation of the First Affiliated Hospital of Chengdu Medical College (CYFY2019ZD04), Natural Science Foundation of Chengdu Medical College (CYZZD21-01) and Sichuan Medical Research Youth Innovation Fund (Q21037)"

4. We note that you have referenced (Costa LE, Boveris A, Koch OR, Taquini AC. Liver and heart mitochondria in rats submitted to chronic hypobaric hypoxia. The American Journal of Physiology. 1988;255(1 Pt 1):C123-9. Epub 1988/07/01. doi: 10.1152/ajpcell.1988.255.1.C123. PubMed PMID: 2839034.) which has currently not yet been accepted for publication. Please remove this from your References and amend this to state in the body of your manuscript: (ie “Bewick et al. [Unpublished]”) as detailed online in our guide for authors

Reviewers' comments:

Reviewer's Responses to Questions

**Comments to the Author**

1. Is the manuscript technically sound, and do the data support the conclusions?

Reviewer #1: Partly

Reviewer #2: Yes

2. Has the statistical analysis been performed appropriately and rigorously? 

Reviewer #1: I Don't Know

Reviewer #2: Yes

3. Have the authors made all data underlying the findings in their manuscript fully available?

Reviewer #1: No

Reviewer #2: Yes

4. Is the manuscript presented in an intelligible fashion and written in standard English?

Reviewer #1: Yes

Reviewer #2: Yes

5. Review Comments to the Author

Reviewer #1: Cyanotic congenital heart disease (CCHD) is characterized by chronic hypoxia. In this manuscript, the authors aim to utilize an in vivo hypoxia cell model to identify potential treatment targets through proteomic analysis. Their findings suggest that chronic hypoxia increases fatty acid metabolism to sustain the growth of cardiomyocytes. However, several concerns regarding data quality and methodology are outlined below:

1. Cardiomyocytes are widely known to preferentially use glucose over fatty acids as energy resources in response to chronic hypoxia (PMID: 17761770; PMID: 33663226). This contradicts the results presented in the manuscript, and the authors need to furnish additional evidence supporting their conclusion, such as conducting glucose and fatty acid uptake and oxidative experiments in response to hypoxia.

2. On page 17, lines 393-394, the statement that "the protein level of CD36 was not changed" does not necessarily indicate unaffected fatty acid uptake since CD36 activation involves translocation from the cytosol to the membrane. To strengthen their findings, the authors should measure membrane CD36 content and fatty acid uptake in response to hypoxia.

3. The proteomics data with identifier PXD042055 is not accessible via ProteomeXchange Datasets, raising a concern about data transparency and accessibility.

4. The choice of mouse cardiomyocytes (MCM) as opposed to the more widely used cardiomyocyte cell lines, H9C2 (derived from rats) and HL1 (derived from mice), necessitates validation. The authors should furnish evidence confirming the identity of the cells used as mouse cardiomyocytes.

5. Principal Component Analysis (PCA) for the proteomic data is crucial for a comprehensive understanding of the dataset and should be provided by the authors.

6. Blast result shows that the shRNA-targeting sequences for ECL1 list in the manuscript is shECI2-targeting sequences, but their western results show knockdown of ECI1, this inconsistency makes their results less reliability.

7. The authors are urged to provide original data, particularly for Western blotting, to ensure transparency and facilitate a thorough evaluation of the results.

Reviewer #2: Full Title: Comparative proteomics reveals that fatty acid metabolism is involved in myocardial adaptation to chronic hypoxic injury Comparative proteomics reveals that fatty acid metabolism is involved in myocardial adaptation to chronic hypoxic injury

The authors have investigated that fatty acid metabolism is involved in myocardial adaptation to chronic hypoxic injury. The manuscript is well addressed and well written. The scientific aptitude of this work is worthy of investigation. The experiments are well designed and executed along with strong evidence of data analysis and statistics.

Comments:

1. The immunoblot experiments done from cardiomyocytes are from whole cell lysates or mitochondrial fraction?

2. Is there any evidence of post transational modification of CPT1A and HIF1-α under chronic and acute hypoxia? Please discuss briefly in the discussion section.

6. PLOS authors have the option to publish the peer review history of their article (what does this mean?). If published, this will include your full peer review and any attached files.

Reviewer #1: No

Reviewer #2: No

---

## [Author Response · Author response to Decision Letter 0]

23 Apr 2024

Response：Thank you for your reminder. Accordingly, we changed the format of the revised manuscript on the basis of PLOS ONE’s style requirements.

2. Thank you for stating the following financial disclosure: "the High-level Talents Research Foundation of the First Affiliated Hospital of Chengdu Medical College (CYFY-GQ09 and CYFY-GQ37), Research Foundation of the First Affiliated Hospital of Chengdu Medical College (CYFY2019ZD04), Natural Science Foundation of Chengdu Medical College (CYZZD21-01) and Sichuan Medical Research Youth Innovation Fund (Q21037)"

Response：We added the statement "The funders had no role in study design, data collection and analysis, decision to publish, or preparation of the manuscript." in revised manuscript.

Response：We provided the original uncropped and unadjusted images in Supporting Information.

4. We note that you have referenced (Costa LE, Boveris A, Koch OR, Taquini AC. Liver and heart mitochondria in rats submitted to chronic hypobaric hypoxia. The American Journal of Physiology. 1988;255(1 Pt 1):C123-9. Epub 1988/07/01. doi: 10.1152/ajpcell.1988.255.1.C123. PubMed PMID: 2839034.) which has currently not yet been accepted for publication. Please remove this from your References and amend this to state in the body of your manuscript: (ie “Bewick et al. [Unpublished]”) as detailed online in our guide for authors http://journals.plos.org/plosone/s/submission-guidelines#loc-reference-style

Response：Thank you for your advices. We deleted this reference in revised manuscript.

Response：Accordingly, we uploaded the Supporting Information files with captions.

---

## [Decision Letter · Decision Letter 1]

6 May 2024

PONE-D-23-43179R1Comparative proteomics reveals that fatty acid metabolism is involved in myocardial adaptation to chronic hypoxic injuryPLOS ONE

Dear Dr. Chen,

Thank you for submitting your manuscript to PLOS ONE. After careful consideration, we feel that it has merit but does not fully meet PLOS ONE’s publication criteria as it currently stands. Therefore, we invite you to submit a revised version of the manuscript that addresses the points raised during the review process.

We look forward to receiving your revised manuscript.

Kind regards,

Partha Mukhopadhyay, Ph.D.

Section Editor

PLOS ONE

Journal Requirements:

**Additional Editor Comments:**

Your manuscript was reviewed by same experts and we received positive feedback. However, one reviewer asked few minor questions, which require your attention during revision.

Reviewers' comments:

Reviewer's Responses to Questions

**Comments to the Author**

1. If the authors have adequately addressed your comments raised in a previous round of review and you feel that this manuscript is now acceptable for publication, you may indicate that here to bypass the “Comments to the Author” section, enter your conflict of interest statement in the “Confidential to Editor” section, and submit your "Accept" recommendation.

Reviewer #1: (No Response)

Reviewer #2: All comments have been addressed

2. Is the manuscript technically sound, and do the data support the conclusions?

Reviewer #1: Partly

Reviewer #2: Yes

3. Has the statistical analysis been performed appropriately and rigorously? 

Reviewer #1: Yes

Reviewer #2: I Don't Know

4. Have the authors made all data underlying the findings in their manuscript fully available?

Reviewer #1: Yes

Reviewer #2: Yes

5. Is the manuscript presented in an intelligible fashion and written in standard English?

Reviewer #1: Yes

Reviewer #2: Yes

6. Review Comments to the Author

Reviewer #1: The authors have revised the manuscript; however, there are still several comments to address, as outlined below:

1. Please provide quantification results for Figure 6C, including membrane CD36, total CD36, and the membrane/total ratio of CD36.

2. Regarding HL1 cells:

a) Please specify the source of the HL1 cells.

b) Properly cultured HL1 cells typically exhibit adherent, fibroblast-like morphology. However, in Figure 1B and Figure 6C, the cell morphology appears rounded. Additionally, the low confluence and apparent suboptimal cell conditions raise concerns.

c) In the Materials and Methods section, the authors stated the use of DMEM to culture HL1 cells. However, optimal culture medium for HL1 cells is Claycomb medium (PMID: 9501201, PMID: 14766671), which contains numerous proteins, growth factors, and hormones crucial for maintaining cardiac myocyte phenotype and metabolism. Could the choice of culture medium have influenced the notable discoveries reported in this manuscript?

3. On page 4, line 88, there is a duplication of "HL1"; one instance should be removed.

4. On page 7, line 192, "Western Blotting" should be corrected to "Immunofluorescent."

5. On page 7, line 201, "MCM" should be revised to "HL1."

Reviewer #2: The authors have addressed the all the comments. The manuscript is now ready to be accepted for publication

7. PLOS authors have the option to publish the peer review history of their article (what does this mean?). If published, this will include your full peer review and any attached files.

Reviewer #1: No

Reviewer #2: No

---

## [Author Response · Author response to Decision Letter 1]

28 May 2024

Dear Partha Mukhopadhyay,

Thank you very much in handling our manuscript (PONE-D-23-43179R1). We very much appreciated the reviewers for the overall comments/constructive suggestions. Accordingly, we have conducted a new set of analyses. These results have been integrated in the revised manuscript. We believe that all the questions raised from the reviewers have been fully addressed. Our point-by-point response is attached. We hope that this revised manuscript is acceptable for publication on “PLOS ONE”. 

in addition, we changed our fouding infromation. The detail is shown in cover letter.

Thank you very much for your time and effort and we look forward to hearing from you.

Sincerely,

Hu Chen, Ph.D.

---

## [Editor Report · Decision Letter 2]

2 Jun 2024

Comparative proteomics reveals that fatty acid metabolism is involved in myocardial adaptation to chronic hypoxic injury

PONE-D-23-43179R2

Dear Dr. Chen,

We’re pleased to inform you that your manuscript has been judged scientifically suitable for publication and will be formally accepted for publication once it meets all outstanding technical requirements.

Kind regards,

Partha Mukhopadhyay, Ph.D.

Section Editor

PLOS ONE